# Mouse CCL9 Chemokine Acts as Tumor Suppressor in a Murine Model of Colon Cancer

**Marzena Łazarczyk [1,\*], Ewa Kurzejamska [2,3], Michel-Edwar Mickael [1], Piotr Poznański [1], Dominik Skiba [1], Mariusz Sacharczuk [1,4], Zbigniew Gaciong [5] and Piotr Religa [6,\*]**

1. Department of Experimental Genomics, Institute of Genetics and Animal Biotechnology, Polish Academy of Sciences, 05-552 Jastrzębiec, Poland
2. Department of Immunology, Genetics and Pathology, Uppsala University, 75 185 Uppsala, Sweden
3. Department of Laboratory Medicine, Division of Pathology, Karolinska Institute, 142 86 Stockolm, Sweden
4. Department of Pharmacodynamics, Centre for Preclinical Research and Technology, Medical University of Warsaw, 02-091 Warsaw, Poland
5. Department of Internal Medicine, Hypertension and Vascular Diseases, Medical University of Warsaw, 02-091 Warsaw, Poland
6. Department of Medicine, Karolinska Institute, 171 76 Stockholm, Sweden
* Correspondence: m.lazarczyk@igbzpan.pl (M.Ł.); piotr.religa@ki.se (P.R.)

**Abstract:** Colorectal cancer is the third most frequently diagnosed cancer in the world. Despite extensive studies and apparent progress in modern strategies for disease control, the treatment options are still not sufficient and effective, mostly due to frequently encountered resistance to immunotherapy of colon cancer patients in common clinical practice. In our study, we aimed to uncover the CCL9 chemokine action employing the murine model of colon cancer to seek new, potential molecular targets that could be promising in the development of colon cancer therapy. Mouse CT26.CL25 colon cancer cell line was used for introducing lentivirus-mediated CCL9 overexpression. The blank control cell line contained an empty vector, while the cell line marked as CCL9+ carried the CCL9-overexpressing vector. Next, cancer cells with empty vector (control) or CCL9-overexpressing cells were injected subcutaneously, and the growing tumors were measured within 2 weeks. Surprisingly, CCL9 contributed to a decline in tumor growth in vivo but had no effect on CT26.CL25 cell proliferation or migration in vitro. Microarray analysis of the collected tumor tissues revealed upregulation of the immune system-related genes in the CCL9 group. Obtained results suggest that CCL9 reveals its anti-proliferative functions by interplay with host immune cells and mediators that were absent in the isolated, in vitro system. Under specific study conditions, we determined unknown features of the murine CCL9 that have so far bee reported to be predominantly pro-oncogenic.

**Keywords:** CCL9; chemokine; CCR1; colon cancer

## 1. Introduction

Colorectal cancer (CRC) types encompass tumors with high microsatellite instability (MSI) caused by a defective DNA mismatch repair (MMR) system, which occurs sporadically in 15% of all CRC patients, as well as tumors with low MSI and an unaffected MMR that account for approximately 85% of CRC cases. Unfortunately, the latter type is highly resistant to currently available immunotherapies of the disease, such as immune checkpoint inhibitors specifically targeted to programmed cell death 1 (PCD-1) or cytotoxic T lymphocyte antigen 4 (CTLA-4) [1]. Signaling molecules, as chemokines, enable the trafficking of immune cells along the concentration gradient to the site of the accumulation of cancer antigens. Chemokines are ligands binding to the recognized chemokine receptors belonging to the G protein-coupled receptor (GPCR) family [2]. The general role of

chemokines in physiology include ensuring proper orientation of the migrating cells during development and morphogenesis [3], directing immature lymphocytes from bone marrow to the thymus and lymph nodes, wound healing, tissue regeneration, and participation in the leukocyte extravasation at sites of tissue damage or infection [4].

Mouse C-C motif ligand 9 (CCL9), alternatively named macrophage inflammatory protein 1γ (MIP-1γ), was identified in 1995 [5] and is homologous to the mouse CCL6, as well as human CCL23 and CCL15 [6,7]. These four molecules belong to the NC6 subfamily gathering chemokines that undergo N-terminally truncation to become fully active with 10–100 fold increase in affinity, predominantly to the CCR1 receptor [8,9]. Cancer research on CCL9 in mouse models revealed the chemokine's dual nature. CCL9 may act as a pro-oncogenic or anti-oncogenic mediator. Using both murine 4T1 breast cancer and B16F10 melanoma models, it has been determined that accumulation of the host immature myeloid cells releasing CCL9 in the lungs were an early indicator of impending colonization and survival of metastatic cells in the organ [10]. Furthermore, the expression level of CCR1—a key receptor for the CCL9 may correlate with a more malignant phenotype. It was confirmed in a study on murine models of melanoma, where less metastatic murine melanoma B16F1 cells displayed a lower level of CCR1 transcription as compared to the more aggressive murine melanoma B16F10 cells [10]. Contrary to the metastasis-supportive ability of CCL9, studies on mouse models of leukemia or renal cell carcinoma revealed the opposite effect. CCL9-overexpressing murine 32D myeloid progenitor cells had reduced leukemogenic potential in C3H/HeJ mice, resulting in substantial inhibition of tumor growth [10]. Similarly, the murine myeloid 32D cell line, when transformed by the chimeric BCR/ABL oncogene, exhibited CCL9 downregulation followed by a remarkably increased ability to cause leukemia in vivo, while restoration of the chemokine expression decreased the leukemogenic potency, likely due to CCL9-mediated activation of CD3+ cells [11]. Furthermore, treatment of renal cell carcinoma-bearing mice with IL-2/anti-CD40 antibody led to an increase in CCL9 levels resulting in anti-cancer and anti-metastatic effects through CCL9-mediated immune cells' attraction to the tumor [12]. Considering two mCCL9 homologs, human CCL23 and CCL15, these chemokines also display distinct functions that depend on, among others, the cancer type and the disease stage. Recently, CCL23 was demonstrated to promote the migration of human ovarian cancer cells in vitro by activating ERK1/2 and PI3K pathways and was hypothesized to be an important mediator of metastatic cell colonization in the omentum in advanced ovarian cancer patients [13]. Another study on patients with acute myeloid leukemia (AML) showed a relationship between high levels of CCL23 and a higher percentage of leukemic cells in peripheral blood [14]. In breast cancer patients, levels of CCL23 and CCR1 expression correlated with metastasis and decreased survival; however, HER2-positive breast cancer cases corresponded with low levels of CCL23/CCR1 expression and better prognosis [10]. Considering the tumor type of interest, CCL23 is significantly downregulated in adenoma and adenocarcinoma tissues derived from colon cancer patients, as compared to normal mucosa; in the study, seven colectomy specimens were analyzed by microarray, each with representative stages of the disease development from normal tissue to polyps and tumors [15]. Further, CCL23 seems to be upregulated in rectal cancer (*n* = 2) as compared to non-rectal tumors (*n* = 4), based on a dot-blot assay (RayBio Biotin Label-based Human Antibody Array I) [16]. The diminished presence of CCL23 in human tumors may suggest a remarkable potential of this chemokine as a therapeutic agent. Another mCCL9 homolog, human CCL15 chemokine, was evidenced to be the most abundantly expressed chemokine in human hepatocellular carcinomas (HCC). The mechanisms of the disease progression included CCL15-mediated autocrine stimulation of the tumor invasion and recruitment of CCR1+ cells, of which 80% were CD14+ monocytes inducing activity of pro-tumor factors and accelerating metastasis [17]. CCL15 has been postulated as a specific proteomic biomarker of HCC due to its increase in patients' serum with hepatocellular carcinomas [18].

Our previous study on a transwell co-culture consisting of mesenchymal stem cells (MSCs) and mouse colon carcinoma CT26.CL25 cancer cells (further referred to as CT26 cell line) revealed upregulation of, among others, ccl7 or ccl9 gene expression in CT26 cells growing in the company of MSCs [19]. Therefore we suspected that CCL9 might be involved in cancer and decided to generate a CCL9-overexpressing version of the cell line and to study the effect of CCL9 abundance on cancer cell proliferation and migration..

## 2. Materials and Methods

### 2.1. Cell Lines

Mouse colon carcinoma CT26.CL25 cell line (further referred to as CT26 cell line) was purchased from American Type Culture Collection (ATCC, Manassas, VA, USA) and used to obtain lentivirus-mediated CCL9 overexpressed-CT26 cancer cells (mCCL9+) or blank control CT26 cells transfected with empty vector, according to the manufacturer's protocol (Applied Biological Materials Inc., Richmond, Canada), as previously described [19]. Cells were cultured in RPMI 1640 medium (ThermoFisher, Waltham, MA, USA) with supplements as previously described [19] according to ATCC recommendations. Transfected cells were selected by adding puromycin to the culture medium. Cultures were maintained at 37 °C in a humidified atmosphere of 5% $CO_2$/95% $O_2$. CCL9 overexpression was confirmed at the protein level in serum-free media collected from growing cultures in vitro using MIP-1 gamma Mouse ELISA Kit (ThermoFisher, Waltham, MA, USA) according to the manufacturer's protocol.

### 2.2. MTS Proliferation Assay

CT26.CL25 cells, blank control, or mCCL9+ CT26 cells were seeded in 96-well plates with a density of 2500 cells/well. In the case of CT26.CL25, mouse MIP-1γ recombinant protein (Applied Biological Materials Inc., Richmond, Canada) at concentrations of 5, 10, or 15 ng/mL was added exogenously to the medium. The proliferation of transfected cell lines was also established (cultured without the addition of recombinant CCL9 protein). In order to confirm that the CCR1 receptoris involved in cancer cell proliferation, a CCR1 antagonist, BX471 compound (Sigma-Aldrich, Saint Louis, MO, USA), was added at different concentrations ranging from 0.1 to 10 μg/mL to the media of growing CT26 cell cultures. MTS test was performed by adding of 20 μL of CellTiter 96® Aqueous One Solution Reagent (Promega Corporation, Fitchburg, WI, USA) to each well and incubating for 1 h at 37 °C. The absorbance was read at a wavelength of 490 nm using a microplate reader (Agilent BioTek, Santa Clara, CA, USA).

### 2.3. Migration Assay

The ability to migrate of unmodified CT26 cells growing in medium from blank control or CCL9-overexpressing CT26 cells was evaluated using a QCM Chemotaxis Cell Migration Kit (Millipore, Burlington, MA, USA) according to the manufacturer's protocol. A transwell system with 8 μm pores was used. Additionally, a scratch assay was performed on modified cell lines. Briefly, the cell cultures of full confluency were scratched using a p200 pipette tip, followed by media aspiration and adding fresh portions after washing the wells with PBS to remove detached cells. Scratched cell monolayers were photographed at 0 and after 24 h and analyzed using Image J software (National Institutes of Health, Bethesda, MD, USA). Ten individual measurements of wound width for each scratch assay were performed and averaged. Each experiment was performed in three replicates.

*2.4. Tumor Growth Assessment*

BALB/ccmdb female mice (strain originated from Charles River), aged 6–8 weeks, were purchased from the Experimental Medicine Center at the Medical University of Bialystok (Bialystok, Poland). The animals were divided into control ($n = 10$) and test groups ($n = 10$) and were implanted on the back of mouse either with blank control or CCL9-overexpressing CT26 cells ($3 \times 10^6$ cells per mouse). Growing tumors were measured with a caliper every 2 days, starting from day 10 post-injection. On day 24, mice were anesthetized, and the primary tumors were cut off. The collected tissues were used for microarray analysis. The study on animals was performed upon approval no. 28/2015, dated 12 May 2015, obtained from the II Local Ethics Committee at the Medical University of Warsaw in Poland.

*2.5. Microarray*

Samples of tumor tissues ($n = 3$ per group) were frozen immediately upon excision. RNA was extracted using RNeasy Mini Kit (Qiagen, Hilden, Germany) according to the manufacturer's protocol. Gene expression profiles of the tumors were analyzed using an Affymetrix (Mouse Gene 2.1 ST array) at the core facility of Karolinska Institutet (Bioinformatics and Expression Analysis, Sweden). A heatmap was created using the online software, Heatmapper (Basel, Switzerland).

*2.6. Statistical Analysis*

Obtained results were analyzed using STATISTICA 13.3 software (TIBCO Software Inc., Palo Alto, CA, USA). In the first step, data were tested for normality with a Shapiro-Wilk test. Data were analyzed with two-way repeated measures ANOVA (RM-ANOVA) with cell line and time as independent factors. In the case of the significant impact of particular factors indicated by RM-ANOVA, the Fisher least significant difference (LSD) post hoc test was calculated for comparisons between particular groups. Data obtained from the migration assay were compared using the T-student test. Graphs were created with GraphPad Prism 5.04 software (GraphPad Software, San Diego, CA, USA) and expressed as mean ± SEM.

## 3. Results

*3.1. Effect of CCL9 or CCR1 Antagonist on CT26 Cell Proliferation In Vitro*

Two-way RM-ANOVA (cell line and time as independent factors) revealed that there were no significant differences in cell proliferation between blank control and CCL9-overexpressing CT26 cells ($p = 0.28$) in the MTS test. Non-significant interaction between time x cell ($p = 0.11$) also confirmed that cells of both modified lines proliferated similarly in time (Figure 1).

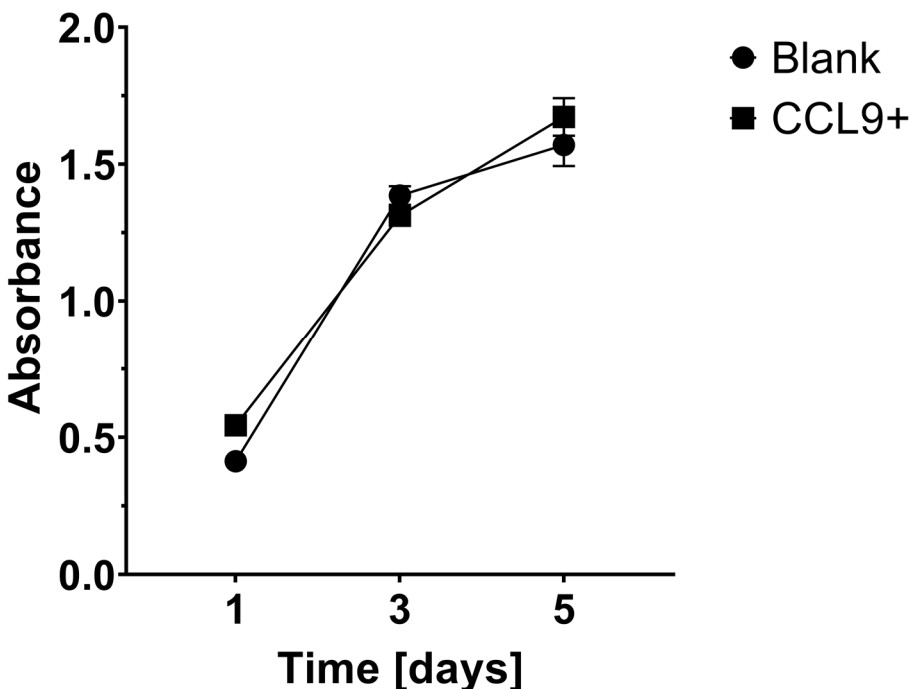

**Figure 1.** Cell proliferation rate assessed in the MTS test revealed no differences between blank control (CT26 cells carrying empty vector) or CCL9-overexpressing CT26 cells. Each experiment was performed in three replicates.

Another MTS test was performed to investigate whether the exogenous addition of recombinant mouse CCL9 to culture affects unmodified CT26.CL25 cell proliferation (Figure 2). Two-way RM-ANOVA (treatment and time as independent factors) showed that supplementation of the cell culture with different doses of CCL9 protein did not affect cell proliferation rate ($p = 0.27$ − treatment; $p = 0.88$ − treatment × time).

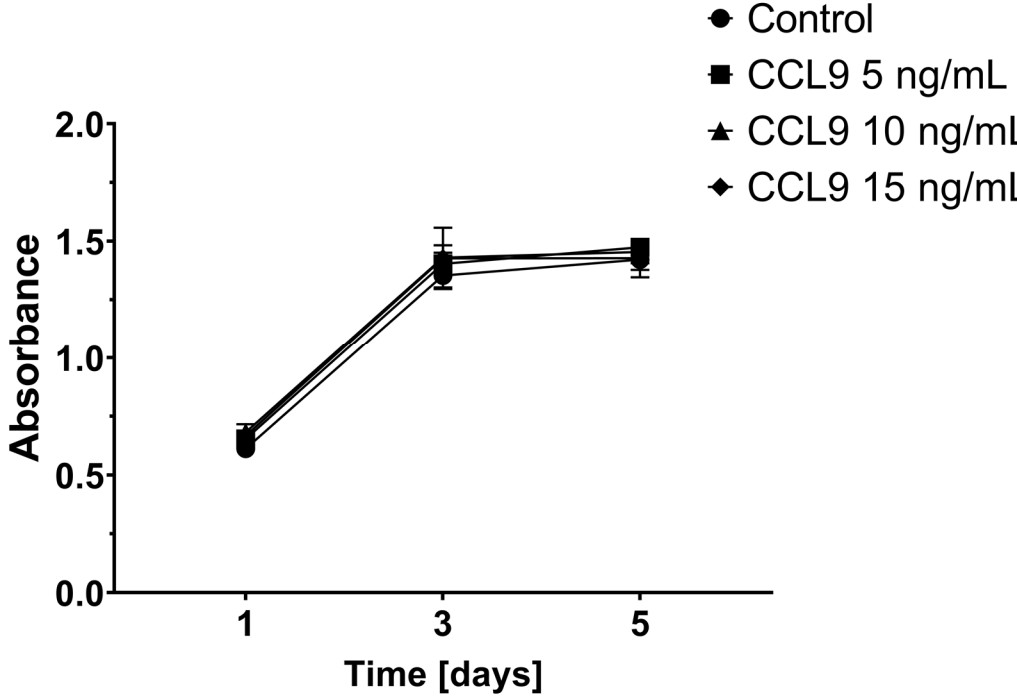

**Figure 2.** CT26 cell proliferation upon CCL9 addition to the culture medium was evaluated by MTS tests. Each experiment was performed in three replicates.

Further proliferation assessment of unmodified CT26 cells utilized three different doses of CCR1 antagonist BX471 to determine the significance of the CCL9-targeted receptors for cancer cell proliferation (Figure 3). Two-way RM-ANOVA (treatment and time as independent factors) showed that blockage of CCR1 caused a significant decrease in cell proliferation rate ($p < 0.001$ − treatment). Moreover, significant time x treatment interaction ($p < 0.001$) revealed that the effect of CCR1 antagonism was more prominent on the 5th day of the experiment. Moreover, two-way RM-ANOVA (dose and time as independent factors) showed that the effect of BX471 was highly dose-dependent ($p < 0.001$ − dose) and confirmed that its effect was also time-dependent ($p < 0.001$ − time x dose interaction).

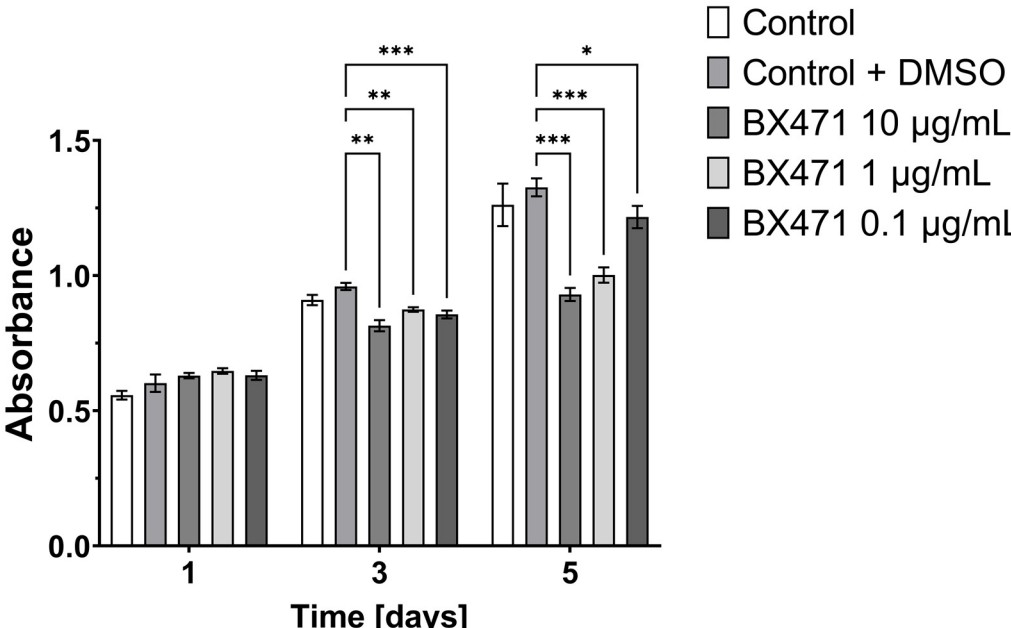

**Figure 3.** Effect of CCR1 receptors blockade by the antagonist BX471 on the CT26 cell proliferation. Each experiment was performed in three replicates. One, two, or three asterisk symbols denote significant results at p < 0.05, p < 0.01, or p < 0.001, respectively.

### 3.2. Effect of CCL9 on CT26 Cell Migration In Vitro

The migration rate of the CT26 cells was assessed by transwell migration assay. Unpaired t-test indicated that there were no differences ($p = 0.08$) in migration rate between CT26 cells growing in the medium collected from blank or CCL9-overexpressing CT26 cells culture (Figure 4).

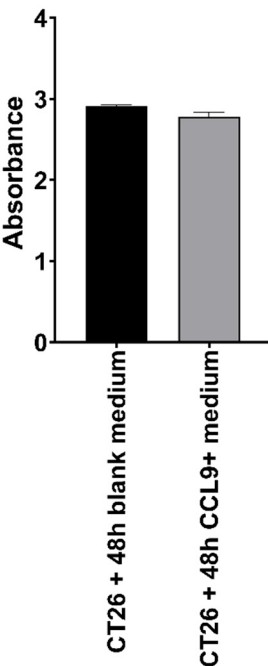

**Figure 4.** Migration of the unmodified CT26 cells upon the influence of the media collected from 48-h cultures of blank control or CCL9-overexpressing CT26 cells.

Migration of the modified cell lines was also assessed in scratch assays (Figure 5). Two-way RM-ANOVA (cell line and time as independent factors) indicated that there were no differences in migration rate between both cell lines ($p = 0.62$ − cell line factor). Moreover, scratch width decreased over time ($p < 0.001$ − time factor), but this process was equal in both cell lines as indicated by non-significant cell line x time interaction ($p = 0.67$).

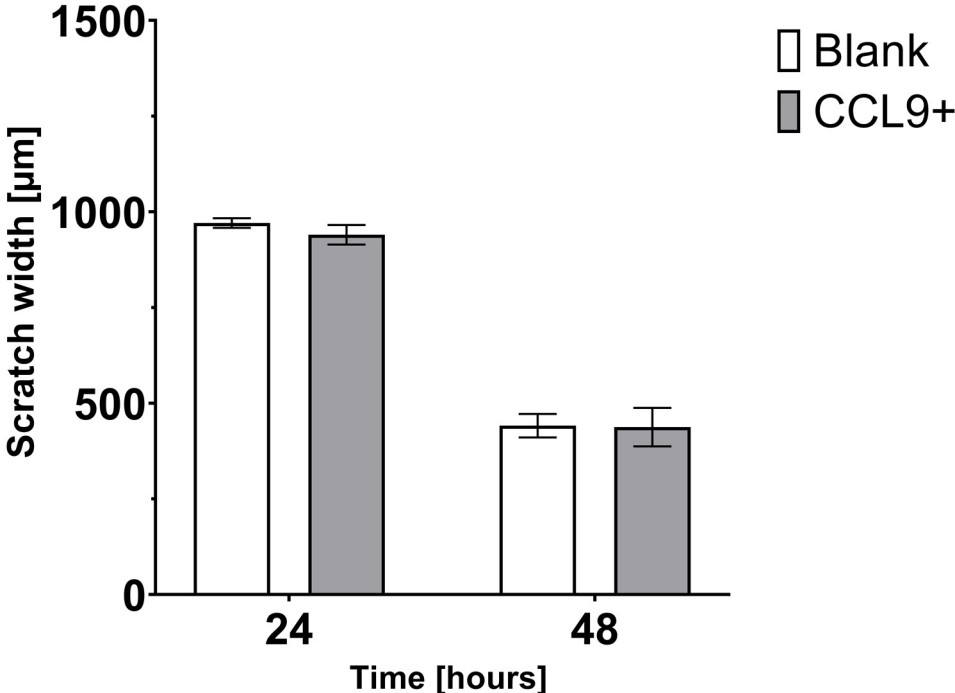

**Figure 5.** Scratch assay evaluating the ability of the modified CT26 cells to migrate.

*3.3. Tumor Growth in BALB/c Mice Inoculated with Blank Control or CCL9-Obverexpressing CT26 Cells*

In the in vivo experiment, we observed a profound decrease in tumor growth in CCL9-overexpressing CT26 tumor-bearing mice, as compared to the control group (Figure 6).

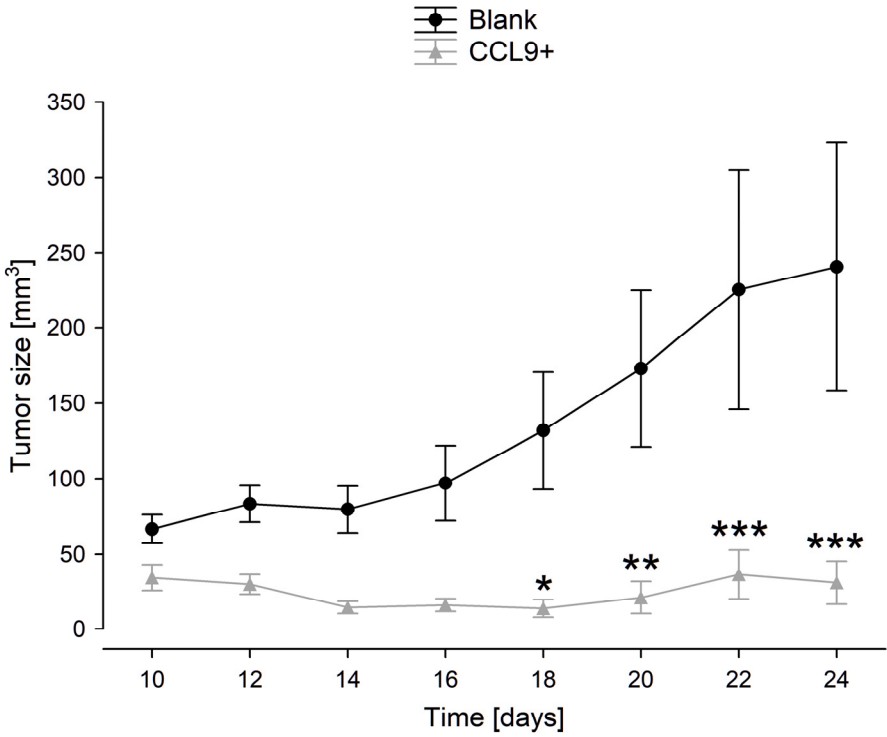

**Figure 6.** Tumor growth in BALB/c mice injected sc. with blank control CT26 cells or CCL9-overexpressing CT26 cellsOne, two, or three asterisk symbols indicate significant results at $p < 0.05$, $p < 0.01$, or $p < 0.001$, respectively.

*3.4. Microarray Results and Bioinformatics Analysis*

Comparing gene expression profiles of control vs. CCL9-overexpressing tumors, we found significant increases in Traj40 (T cell receptor alpha joining 40) and Trbv13-2 (T cell receptor beta, variable 13-2) expression levels in tumors with CCL9 upregulation (Figure 7).

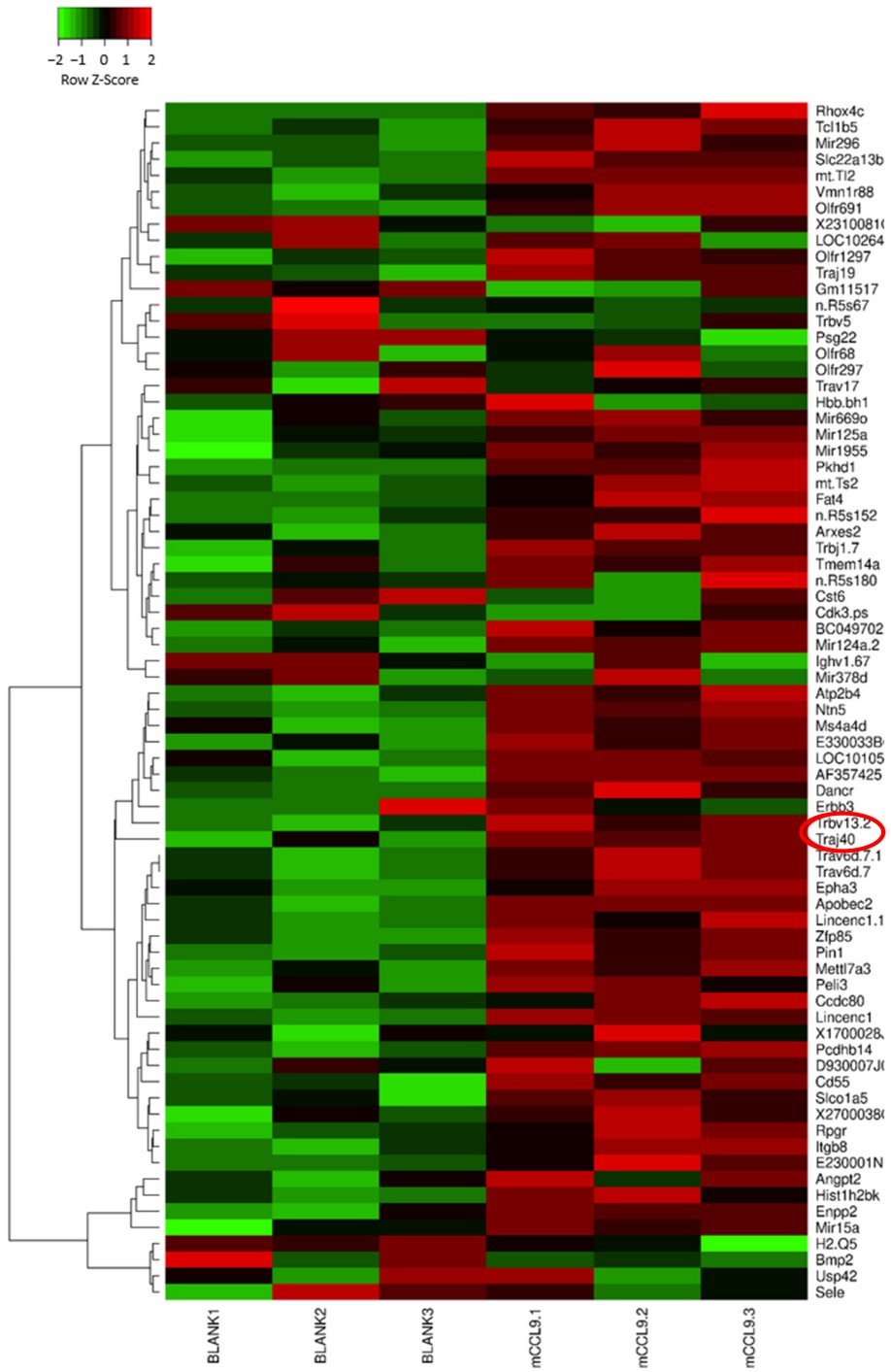

**Figure 7.** Heatmap with up- or downregulated genes in collected tumors. Differences in values between control and mCCL9+ groups were assessed by *t*-test, where $p < 0.05$ was considered statistically significant.

Both Traj40 and Trbv13-2 code for components of alpha and beta polypeptide chains forming T cell receptors (TCR) in RORγt CD4+ T cells, therefore the results indicate a correlation between CCL9 and RORγt CD4+ T cell activation (Supplementary Figure S1A–C). Bioinformatic analysis of ChIP-seq data from a study by Ciofani et al. for Th17 cells revealed that RoRc directly controls the expression of Traj40 [20]. Furthermore, we analyzed a public dataset for human colorectal cancer by comparing microarray data from cancerous versus non-cancerous regions in the column (geo GSE136735). We found a correlation

between RORγt expression, TCR genes, and CCL23 (human counterpart of murine CCL9) (Figure 8A).

Interestingly, our microarray results also showed that interleukin 6 (IL6) is upregulated in the tumors resulting from the implantation of CCL9-overexpressing CT26 cells (please refer to the Figshare repository: https://doi.org/10.6084/m9.figshare.13713817).. IL6 production has been shown to support Th17 cell differentiation [21-23]. Interleukin 6 is known to be produced by activated immune cells and stromal cells, including T cells, monocytes/macrophages, endothelial cells, fibroblasts, and hepatocytes [24]. We analyzed the GEO RNA-seq data given by GSE50760 from 18 patients with diverse progression and heterogeneity of CRCs using BioJupies notebook [25]. Our data confirm the upregulation of IL6 in CRC (Supplementary Figure S2). To gain a deeper insight into IL6 expression level in colorectal cancer cells, we analyzed a scRNA-seq dataset collected from a colorectal patient available publicly using the GEO repository (ID: GSE222300) through the use of a Seurat pipeline (Supplementary Figure S1D). Our analysis shows that RORc+ cells cluster with other CD4+ and CD8+ phenotypes, including Th1 and Th22 (Supplementary Figures S1E and S1F). It also shows that IL6 is not predominantly expressed by this cluster (i.e., cluster 3) (Supplementary Figure S1G). Additionally, it shows that CCL23 and IL6R are significantly downregulated, and RORc is slightly downregulated, thus confirming our other investigation results (Supplementary Figures S2A and S2B). These results suggest that the recruitment process and the release of IL6 produced by the recruiting cells could be at least in part secluded from the recruited cells (in this case, Th17 cells). This could be due to the effort of regulatory immune cells (e.g., Treg cells, MDSCs, M2 macrophages, and cancer-associated fibroblasts) that produce anti-inflammatory factors to suppress anti-tumor immune cells [26].

The reasoning behind IFNγ expression could be related to the phenotype-based function of Th17 cells (Figure 8a). To investigate the reason behind this observation, we searched the human protein atlas for IFNγ expression (Supplementary Figure S3). Our analysis indicates that IFNγ-expressing cells include natural killer cells and granulocytes. IFNγ is also known to be produced by, among others, Th1 and NK cells. This observation raises the possibility that IFNγ production could be independent of Th17 status. Furthermore, another feasible explanation could be related to the phenotype-based function of Th17 cells. Colorectal cancer-derived Th17 triggered the release of pro-tumorigenic factors by tumor and tumor-associated stroma. However, the same cells increased the recruitment of neutrophils through interleukin 8 (IL-8) secretion and attracted cytotoxic CCR5+CCR6+CD8+T cells into tumor tissue through CCL5 and CCL20 production [27]. Additionally, Th17 cells have several phenotypes, including Th17(Conventional) and Th17(Th1-like). There are differences between Th17(Conventional) and Th17(Th1-like) related to their differentiation condition. Th17(Conventional) cells differentiation takes place in the presence of transforming growth factor β1 (TGFβ1) and IL6, while Th17(Th1-like) can occur upon the influence of the interleukins: IL1β, IL6, and IL23, as well as IL1β+ IL6 alone or TGFβ3+IL6+IL23 condition [28]. Another difference between Th17(typical) and Th17(Th1-like) is the production of pro-inflammatory cytokines [29]. Whereas, Th17 is known for the production of IL17A and IL17F. Th17(Th1-like) cells can produce IFNγ and Tbet [30].

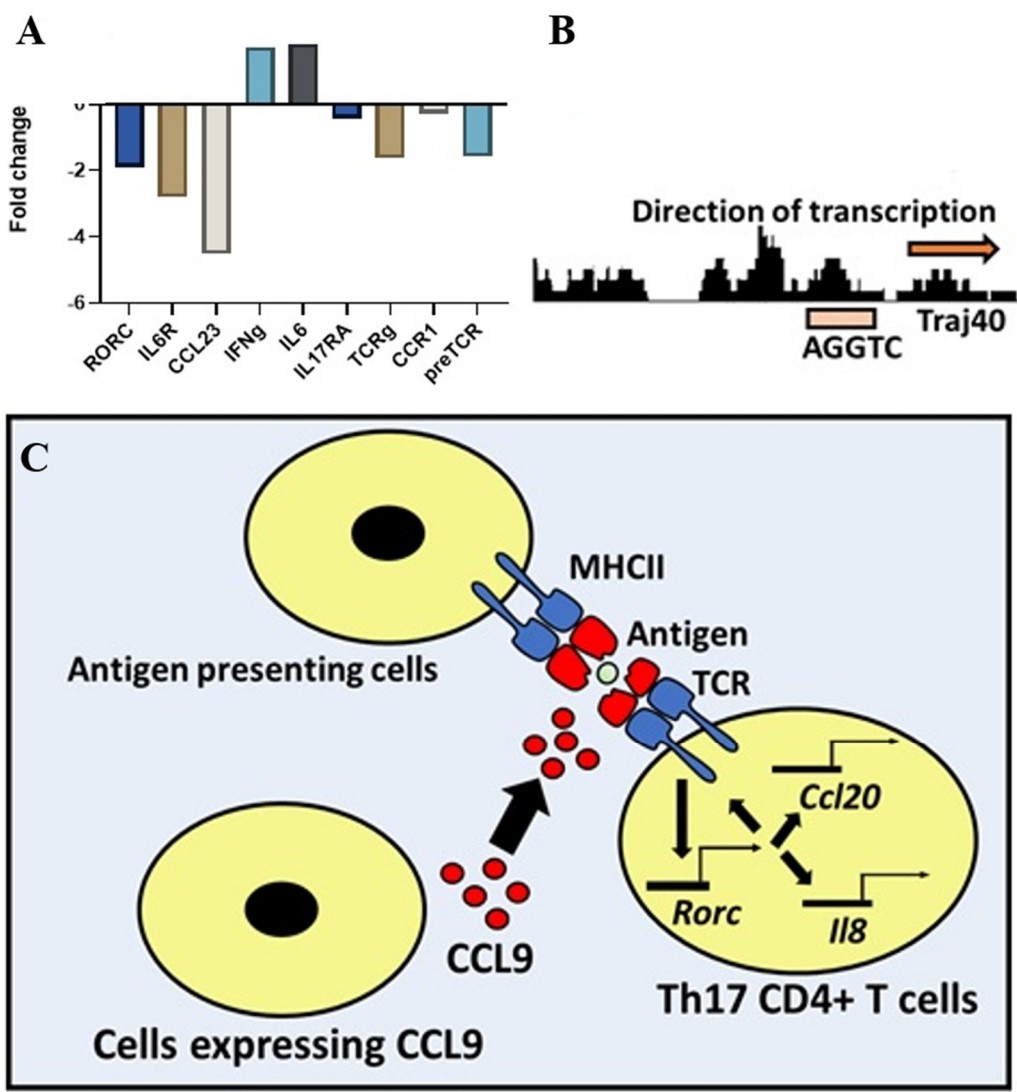

**Figure 8.** Microarray-based analysis suggests the involvement of CCL9 in CD4+ Th17 cells recruitment/activation in colon cancer. (**A**) Human microarray (GSE136735 on GEO repository) analysis of colorectal tissue versus normal tissues indicated downregulation of RORγt, CCL23, IL17RA, and TCRG (TARP) gene values in colorectal cancer samples, as shown in the bar plot. Interestingly, CCR1 (i.e., CCL23 receptor in humans) expression was not significantly altered in colon cancer tissue as compared to normal tissue, suggesting that the CCL9 effect on cancer growth might not be CCR1-mediated; (**B**) our analysis of ChIP seq-data published in Ciofani et al. identified RORγt response element motif AGGTCA in the promoter of Traj40 in Th17 cells [20]. This observation indicates that there might be an auto-regulation loop between RORγt and the genes controlling TCR receptor function/diversification. (**C**) The data hints at a possible role for CCL9 in the activation/recruitment of pro-inflammatory Th17 cells, known to be beneficial in slowing tumor growth. A seemingly feasible hypothesis of CCL9 function could be that it plays a role in increasing TCR activation in RORγt+ T cells (i.e., Th17 cells) through a Gq-mediated pathway. In turn, Th17 cells could promote CD8 and neutrophil recruitment, thereby decreasing tumor growth. However, our data do not exclude a positive effect of CCL9 on other RORγt+ T cells, such as RORγt+ CD8 (also known as TC17).

## 4. Discussion

We hypothesize that the anti-tumorigenic effect of CCL9 in our animal study was a result of the chemoattraction of anti-cancer leukocytes into the microenvironment of the CCL9-overexpressing tumors or not yet uncovered the mechanism of the CCL9-mediated tumor suppression; however, it remains to be confirmed. Based on our microarray data,

we assumed that CCL9 could stimulate Th17 cell activity and migration. Therefore, CCL9-based therapy against cancer seems to be worth consideration in further research. Our study revealed profound tumor growth-inhibiting properties of the CCL9 chemokine in a mouse model of colon cancer; however, most reports describe the pro-oncogenic effect of CCL9. For instance, a study on mutant cis-Apc/Smad4 mice spontaneously developing adenocarcinoma of the intestine demonstrated two-fold higher expression of Ccl9 mRNA in polyps than in wild-type C57BL/6, as well as five-fold higher expression of CCL9 protein in polyps than of Apc+/D716 mice [6]. Furthermore, an increased level of CCL9 was associated with enhanced recruitment of CD34+ immature myeloid cells (iMCs) from bone marrow to the tumor site. These iMCs displayed CCR1 expression enabling migration along with CCL9 gradient, as well as intense production of matrix metalloproteinases MMP9 and MMP2; therefore, the iMCs promoted tumor invasion [6]. Another example is the study on a mouse model of KRasG12D-driven lung adenoma that evidenced CCL9-mediated recruitment of macrophages and PD-L1-dependent discrimination of T and B cells, where co-blockade of both CCL9 and IL-23 abolished Myc-induced tumor progression, that confirmed tumorigenic properties of CCL9 [31]. We hypothesize that the anti-cancer effect of CCL9 observed in our in vivo research was a result of mostly anti-cancer leukocyte infiltration of the tumor or a thus far unknown CCL9-mediated mechanism preventing tumor progression beyond CCR1 signaling. Our gene profiling data from collected tumors allow us to postulate that CCL9 could likely regulate Th17 cell activity and migration. It is well-documented that Th17 cells may play a distinct role in colon cancer development. First, Th17 cells are known to induce the release of pro-tumorigenic factors by tumor-associated stroma. Inversely, they were also shown to promote the recruitment of neutrophils through the production of IL-8. Furthermore, Th17 cells were found to be responsible for CCL5 and CCL20 secretion, which are known to recruit CD8+ T cells (CCR5+ CCR6-) in the intraepithelial regions of the tumor [27,32]. As we determined, CCR1 receptors are usually downregulated in human colon cancer tissues, which likely constitutes a tumor escape mechanism from host defense since mCCL9/hCCL23 may possibly act as a chemoattractant for Th17 cells during the process of antigen recognition. Therefore, overexpression of CCL9 could promote Th17 activation and migration, which in turn, supports the recruitment of CD8+ cells and neutrophils. CCL9 was found to be overexpressed in lymph nodes but not within tumors in mice inoculated with breast adenocarcinoma SB5b cells suggesting that the restoration of CCL9 expression within the tumor may result in its growth inhibition [33]. Interestingly, upregulation of the CCR1 receptor instead of CCL9 could be insufficient to decrease tumor cell survival and migration [10]. Leukocytes infiltrating the tissue upon recruitment, along with immune cells naturally inhabiting the skin, provide dynamic balance. Chronic inflammation resulting from growing tumors leads to homeostasis disturbances [34]. In our study, induction of a local increase in CCL9 released from cancer cells injected in the skin could attract the immune cells to the site of a growing tumor, resulting in a reduction of its further enlargement; however, it requires confirmation. Of note, the mouse model of colon cancer we used in the study has been utilized previously [19,35–37]. Although CCL9 emerges with both pro- and anti-cancer properties, which are likely dependent on the specific microenvironment of a given tumor type, available research on human homolog CCL15 in colon cancer models remains in contradiction with our data if we assume that our results are somehow translational into human studies. One example is a study on metastatic samples from colorectal patients that reported the examined tissues contained 3-fold more CCR1(+) cells when expressed CCL15 upon SMAD4 downregulation, which correlated with significantly shorter disease-free survival of the donors [38]. In livers of nude mice, SMAD4-deficient human CRC cells (AA/C1, HT29, Colo205, LoVo, DLD-1, and HCT116) displayed CCL15 upregulation and increase in CCR1(+) cells recruitment that promoted tumor invasion [38]. In our study, microarray data indicated that SMAD4 expression in the tumors was unchanged. Furthermore, the amino acid sequence identities between mouse CCL9

and human homologs are only partial [10]. Therefore, some distinct functionalities cannot be entirely excluded.

### 4.1. Perspectives for CCL9 Use in Anti-Cancer Therapy

So far, the therapeutic effect of mouse CCL9 has only been evidenced in a few mouse models of cancer [11,12]. Considering the therapeutic potential of chemokine CCL9 or other chemokines that were evidenced to contribute to the elimination of tumor cells, the local delivery of these chemoattractants for directed trafficking of the host T cells expressing specific chemokine receptors seems to be the most challenging. Several approaches have been tested in the past that made possible utilization of the natural properties of chemokines to the therapeutic advantage in cancer. Among them are chemokine gene transduction into tumor cells, intratumor injection of chemokine-expressing viral vectors, including adenoviruses, application of transduced dendritic cells (DCs) producing chemokines that induced cytotoxic T cells, attenuated or non-pathogenic microorganisms manipulated to release chemokines, as well combined therapy such as simultaneous delivery of chemotherapeutic drug and viral vectors encoding for chemokines [39]. It seems that delivery via mentioned carriers of more than one distinct chemokine demonstrating different anti-cancer functions could also be a beneficial approach. For example, within the CXC chemokine family, there are members containing ELR motif (Glu-Leu-Arg sequence), and they were demonstrated to be pro-angiogenic, while those lacking the specific amino acid sequence displayed an inhibitory effect on angiogenesis [39]. Therefore, a synchronized increase in the intratumoral level of CCL9 and the other anti-angiogenic chemokine (or administration of, e.g., anti-VEGF drugs like aflibercept, ranibizumab, bevacizumab) would enable the management of more than one process, leading to cancer progression. Recently, increasing attention has been paid to the role of the microbiome in different pathologies. Since the tumor microenvironment is a residence of specific microbiota, located mostly intracellularly (both in tumor and immune cells) [40], bacteria can be engineered to produce therapeutic molecules and used as a drug-delivery platform. Previous attempts assumed, among others, the use of an attenuated strain of *Salmonella typhimurium* engineered to produce potent angiostatic CCL21 and administered systemically (intravenously) to mice injected subcutaneously with CT26 cells [41]. Histological analysis of distinct organs revealed that the modified bacteria accumulated mainly within the primary lesion or metastases and significantly inhibited tumor progression without serious adverse effects. Depletion of CD4+ and/or CD8+ immune cells in mice using specific antibodies confirmed that the inhibitory effect of CCL21 on tumor growth was associated with specific T cell chemoattraction [41]. An improved method of delivery to the tumor of the modified *S. typhimurium* expressing mouse CCL21, avoiding systemic distribution, was reported by Din and collaborators [42]. The bacteria were engineered to undergo repeatedly synchronized lysis circuit (SLC), genetically programmed to switch bacteria self-destruction upon achieving the quorum threshold of the lysis inducer. Lysed bacteria successively released the content; the small bacteria fraction that survived constituted seeds regenerating the population that again colonized the tumor. The approach provided constitutive production of therapy without the necessity of performing multiple injections. The study utilized a subcutaneous model of colorectal cancer (MC26 cell line) in immunocompetent mice, where pulsatile bacterial population dynamics within the tumor was evident, while constant "self-delivery" of CCL21 led to the recruitment of T cells and dendritic cells resulted in a significant reduction in tumor growth [42]. A similar approach has been used very recently, where an engineered probiotic strain of *Escherichia coli* injected into subcutaneously growing tumors in mice (following sc. inoculation of murine A20 B cell lymphoma cells) contributed to the local SLC-mediated release of human CXCL16 and CCL20 [43]. As a consequence, the accumulating chemokines promoted the recruitment of activated T cells and dendritic cells, respectively, which caused tumor re-

gression [43]. The aforementioned examples of loading the tumor with chemokines in living organisms can be translated into the use of mouse CCL9 or human equivalents to develop new anti-cancer strategies.

### 4.2. Limitation of the Study

We are aware of the limitations of our study. First, the human equivalent of mouse CCL9 is only partially consistent in sequence; therefore, the results may not be replicable in studies on clinical material. Second, the key genes identified in microarray analysis require confirmation at the protein level, as well as types of immune cells infiltrating tumors of both experimental groups should be determined. We are planned to continue the research on CCL9 to further validate so far received results.

## 5. Conclusions

Undoubtedly, further studies are required to confirm the CCL9-mediated mechanism of tumor growth inhibition and to verify whether animal research on CCL9 could be translated into clinical use. Our results indicate that there is a relationship between CCL9 and RORγt CD4+ T cell activation and migration. Uncovering that CCL9 stimulation of immune cells almost completely eliminates tumor growth is promising and worthy of further exploration toward the identification of new molecules or molecular targets for colon cancer therapy.

**Supplementary Materials:** The following supporting information can be downloaded at: https://www.mdpi.com/article/10.3390/cimb45040226/s1, Figure S1: Figure S1: A bioinformatics analysis suggesting CCL9 may play a role in the recruitment of RORγt+ T cells.; Figure S2: Bioinformatics analysis of IL6 expression in colorectal cancer.; Figure S3: IFNγ expression by different cell types.. References [44,45] are cited in the supplementary materials

**Author Contributions:** Conceptualization, P.R. and Z.G.; methodology, E.K. and M.Ł.; formal analysis, M.S., P.P., M.Ł., M.-E.M. and D.S.; investigation, M.Ł., P.P., E.K., and M.-E.M.; writing—original draft preparation, M.Ł. and M.-E.M.; supervision, P.R. and Z.G.; funding acquisition, P.R., Z.G., D.S. and M.S. All authors have read and agreed to the published version of the manuscript.

**Funding:** This research received no external funding.

**Institutional Review Board Statement:** The animal study protocol was approved by the II Local Ethics Committee for the Experiments on Animals at the Medical University of Warsaw; 61, Żwirki i Wigury Str., 02-091 Warsaw (permission no.: 28/2015 dated 12 May 2015).

**Data Availability Statement:** The microarray results presented in this study can be found in the online Figshare repository: https://doi.org/10.6084/m9.figshare.13713817 accessed on 15 February 2021.

**Conflicts of Interest:** The authors declare no conflict of interest.

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
