# Peer review of "Mouse CCL9 Chemokine Acts as Tumor Suppressor in a Murine Model of Colon Cancer"

_cimb, doi:10.3390/cimb45040226_

Round 1

Reviewer 1 Report

The present study by Łazarczyk M et al. is very well designed and presented and demonstrates the significance and potential effect of CCL9 in colorectal cancer. Although it may not be 100% translatable to humans, it still provides evidence and proof of concept for the mechanisms. I would suggest authors include a paragraph describing the therapeutic effect of CCL9 and the potential synergetic development of immunotherapies and antiangiogenic therapies in the discussion section.

Author Response

Dear Reviewer

Thank you for your comments. We appreciate it. 

Kind regards,

- authors of the manuscript

Reviewer 2 Report

This manuscript is very well written. The authors provided sufficient scientific evidence for their work. The introduction, material and methods, results, discussion, and conclusion are well described. The authors are well aware of the limitation of their study. The reviewer is satisfied with the work and suggests the manuscript for publication. 

Author Response

(The authors gave the same response as above.)

Reviewer 3 Report

The comments are in the pdf

Author Response

(The authors gave the same response as above.)
